# Prevalence of social anxiety disorder and its associated factors among foreign-born undergraduate students in Türkiye: A cross-sectional study

**Lujain Alnemr**[1], **Abdelaziz H. Salama**[1,2]*, **Salma Abdelrazek**[1‡], **Hussein Alfakeer**[1‡], **Mohamed Ali Alkhateeb**[1], **Perihan Torun**[3]

**1** Hamidiye International School of Medicine, University of Health Sciences, Istanbul, Türkiye, **2** School of Medicine, Johns Hopkins University, Baltimore, Maryland, United States of America, **3** Department of Public Health, Hamidiye International School of Medicine, University of Health Sciences, Istanbul, Türkiye

☯ These authors contributed equally to this work.
‡ SA and HA also contributed equally to this work.
* asalama5@jh.edu, dr.abdelazizsalama@gmail.com

**Data Availability Statement:** in paper and SI - All relevant data are available from within the

## Abstract

Social anxiety disorder (SAD) is prevalent among university students, yet data on its severity among foreign-born international undergraduate students in Türkiye remains limited. This study aims to determine the prevalence of SAD and its associated factors within this population. A cross-sectional study was conducted using a Google Form survey distributed across various universities from September 17, 2023, to February 1, 2024. The survey comprised two sections: sociodemographic information and 17 items of the Social Phobia Inventory (SPIN), which measures the frequency and intensity of social anxiety symptoms. Data analysis included descriptive statistics and inferential analysis, multiple regression, and binomial logistic regression. Out of 506 participants, 455 were included in the study. Results revealed that 39.1% exhibited no or very mild symptoms of SAD, while 23.7% experienced mild symptoms, 21.3% faced moderate symptoms, and 11.6% and 4.2% presented with severe to very severe symptoms, respectively. Factors such as gender (p < 0.0001), previous academic failures (p = 0.013), family history of mental health issues (p = 0.009), exercise frequency (p < 0.0001), and perceptions of relationships with classmates (p < 0.0001) were significantly associated with SAD. Females showed a higher probability of SAD compared to males (OR = 1.976). Individuals engaging in over 90 minutes of exercise per week were less likely to have SAD (OR = 0.383), and occasional smokers had a lower risk of SAD compared to non-smokers (OR = 0.422). Our study uncovered a notably elevated prevalence of Social Anxiety Disorder (SAD) among foreign-born undergraduate students in Türkiye. Factors such as being female, having a family history of mental illnesses, studying in a stressful environment, experiencing academic failure, and engaging in less frequent exercise were associated with noticeable symptoms of SAD. These findings emphasize the urgent need for heightened efforts in recognizing and addressing SAD within this population.

manuscript as well as a supplemental information file.

**Funding:** The authors received no specific funding for this work.

**Competing interests:** The authors have declared that no competing interests exist.

## Introduction

Anxiety disorders are the most common group of mental illnesses in the world and the ninth most health-related cause of disability, according to the World Health Organization (WHO) [1, 2]. Social Anxiety Disorder (SAD), also referred to as social phobia, is classified according to the Diagnostic and Statistical Manual of Mental Disorders V (DSM-V) as an anxiety disorder that presents with an intense fear of being in social situations where one may be under scrutiny by others [3]. SAD usually starts during late childhood and continues throughout life [4, 5]. People with SAD endure severe emotional and/or physical symptoms and may also show signs of shyness and discomfort when in social situations, even though they crave other people's company [6].

Long-term effects of SAD include low self-esteem, decreased quality of life, and academic underachievement [7–9]. Among university students, the prevalence of SAD is very prominent [10–14]. Moreover, enrolling in a university forces students to deal with new life experiences, such as adjusting to new life circumstances, engaging in university society, and fulfilling the demands of life. These factors are recognized to impact students' mental health and their participation in risky behaviors affecting their health, such as smoking, drinking alcohol, eating poorly, or being physically inactive [15].

Students seeking education abroad face significant difficulties adjusting due to the large variations in social and cultural norms [16]. They may experience emotional issues, including loss, separation, travel stress, social pressure, exacerbation of any underlying or hidden disorders, pharmaceutical complications, and unexpected situations [17, 18]. These challenges may, therefore, result in anxiety, anorexia, and general avoidance of social interactions [19, 20].

In a study done on Chinese international students, the researchers found that students faced challenges regarding language, academics, psycho-socio-cultural matters, finances, and political issues which had a negative impact on life satisfaction among the students [21]. Foreign-born students were found to face many difficulties in language and communication, and they can experience isolation and discrimination [22]. These results underline the importance of more research on the mental health of foreign students. Although studies worldwide assessed the prevalence of SAD in different populations, they did not give enough attention to population diversity, nor did they differentiate between foreign-born and non-foreign-born students.

The population of international students pursuing university education in Türkiye is steadily increasing. According to the latest update from the Council of Higher Education (YÖK) in 2023, it is estimated that there are 260,316 international undergraduate students in the country [23]. This indicates that Türkiye provides an excellent opportunity to reach a respectable number of foreign students from different backgrounds. Previous studies highlighted that SAD is already prevalent among Turkish university students [24, 25]. Furthermore, emotional, academic, and financial difficulties were determined to be the most prevalent concerns faced by Turkish students [26]. Another study conducted in 2015 showed that particularly international students in Türkiye commonly experience psychosocial challenges such as being separated from their families, fear of making mistakes, feeling disappointed in their choice to study in Türkiye, social pressure, difficulty expressing thoughts, and struggling with criticism [27].

These various obstacles that international students face make them susceptible to mental health disorders. However, to our knowledge, no study has looked into the prevalence of SAD in this population. This study, therefore, aims to investigate the prevalence of SAD among foreign-born undergraduate students in Turkish universities and its associated factors.

## Methods

### Study design

This study employs a cross-sectional design, utilizing a self-administered questionnaire distributed to foreign-born undergraduate students in Türkiye via Google Forms. To reduce the possibility of non-response bias, we required that all questions be answered before submission. The survey, designed to maximize participation and guarantee anonymity, was conducted from September 17, 2023, to February 1, 2024. As there is no formal list representing international students in Turkish universities, we tried to reach international students from various universities through WhatsApp, Facebook, and Telegram groups that connect international students. Administrators of these groups were requested to send periodic reminders to enhance response rates. Additionally, we approached associations concerned with international students' affairs to promote the survey and send reminders about it.

### Sample size

Based on the most recent data from the Council of Higher Education (YÖK), in 2023, there were 260,316 non-Turkish undergraduate students in Türkiye [23]. Utilizing EpiInfo 7.2.5.0, we determined the sample size required to represent this population. Considering a population size of 260,316, with p = 0.21 and q = 0.79, along with a 95% confidence interval and a 5% margin of error, we calculated a minimum required sample size of 255 participants.

### Participants and survey

The study included undergraduate students who are foreign-born non-Turkish, aged 18 or older and currently enrolled in Turkish universities. The survey is structured into two sections: the first part collects basic demographic and socioeconomic information, presented in both Arabic and English. The second part utilizes the Social Phobia Inventory (SPIN), a validated tool for evaluating symptoms of social anxiety and phobia. The SPIN is administered in both its original English version and its validated Arabic translation, ensuring an accurate assessment of social anxiety symptoms among participants. The questionnaire was restricted to Arabic and English. This choice was driven by the notable representation of Arabic-speaking students in Türkiye, with five of the top 10 international student nationalities in Türkiye being Arabic-speaking [28]. Additionally, the decision was influenced by the assumption that many students seeking education outside their home countries have sufficient English language skills to answer the questionnaire.

SPIN, with its 17 questions, measures the frequency and intensity of social anxiety symptoms over the preceding week. Responses are captured on a 5-point Likert scale from 0 (not at all) to 4 (extremely), allowing for a comprehensive symptom severity score ranging from 0 to 68. The scoring system classifies severity into five categories: none or very mild (less than 20), mild (21–30), moderate (31–40), severe (41–50), and very severe (51 or more), enabling detailed analysis of social anxiety levels among the surveyed population.

### Quality of measure

The Social Phobia Inventory (SPIN), originally developed in English by Connor et al. (2000) [29], has undergone successful translation and validation into Arabic by Sfeir et al. (2022) [30]. The Arabic version has demonstrated high reliability and validity for evaluating social anxiety symptoms among Arabic-speaking populations. In two separate ways, the Arabic SPIN exhibited Cronbach's alpha values of 0.92 and 0.90, indicating strong internal consistency comparable to the original scale. Moreover, its convergent validity and gender-invariant measurement

of social phobia further confirm its efficacy. Consequently, the Arabic SPIN emerges as a robust tool for detecting social phobia in Arabic-speaking individuals.

## Statistical analysis

The statistical analysis of this study encompasses descriptive, inferential analysis, multiple regression, and logistic regression. Initially, descriptive statistics, including frequencies and percentages, will outline the participant characteristics of sociodemographic and academic data. The Shapiro-Wilk test assesses the normality of the Social Phobia Inventory (SPIN) scores. For non-normally distributed data, the Mann-Whitney U and Kruskal-Wallis H tests will examine differences in SPIN scores across demographic variables. Associations between demographic factors and SPIN categories (ranging from "None or Very Mild" to "Very Severe") will be explored using the non-parametric chi-square test.

Further, multiple regression analysis will predict SPIN scores (Ranging from "0" to "68") based on variables such as gender, smoking status, previous failures in university, family history of mental health issues, exercise habits, and classmate relationships. Additionally, a binomial logistic regression will evaluate the impact of various demographic factors, including gender, monthly budget, smoking status, field of study, GPA, family history, Turkish language proficiency, living arrangements, exercise habits, and classmate relationships, on the presence of social phobia, dichotomized into "Absent" (No or very mild SPIN) and "Present" (Mild to very severe SPIN). All statistical analyses will be conducted using IBM's SPSS Statistics package for Windows (Version 25.0).

## Ethical approval

Prior to commencing the cross-sectional study, the research team obtained ethical approval from the Hamidiye Scientific Research Ethics Committee (approval number 23/379) [31]. Participants were briefed on the study's aims and objectives, and digital consent was obtained from them before they completed the survey. Consent was indicated by participants selecting a designated checkbox.

## Results

### Participant characteristics

We initially collected 506 responses but narrowed them down to 455 eligible participants after excluding those who did not meet the undergraduate criterion (37 respondents), one due to Turkish nationality, and 13 for incomplete surveys. The majority of participants were Syrian students (59.6%); second to them were Egyptian students at 11.4%. Gender distribution was nearly balanced (47.9% females, 52.1% males), and over half were aged 18–21 (52.1%). Most were single (94.1%), reported an average monthly budget (48.1%), and were majoring in medical sciences (45.9%). The social classmate environment was predominantly described as friendly and relaxed (56.3%). The significant majority reported being non-smokers (85.7%) and abstaining from alcohol (97.6%). Living arrangements varied, with 39.6% residing with their parents (**Table 1**).

### Prevalence of social phobia

The study reveals the prevalence of social phobia among participants as follows: 39.1% exhibited "no or very mild symptoms," 23.7% experienced "mild symptoms," 21.3% faced "moderate symptoms," while "Severe" to "very severe symptoms" were present in 11.6% and 4.2% of

**Table 1. Participants characteristics.**

| | | n (%) |
|---|---|---|
| | | 455(100) |
| **Age** | 18–21 | 237(52.1) |
| | 22–25 | 181(39.8) |
| | 30–34 | 27(5.9) |
| | 35 or above | 10(2.2) |
| | Total | 455(100 |
| **Gender** | Female | 218(47.9) |
| | Male | 237(52.1) |
| | Total | 455(100) |
| **Marital status** | Single | 428(94.1) |
| | Married | 24(5.3) |
| | Widowed | 1(0.2) |
| | Divorced | 2(0.4) |
| | Total | 455(100) |
| **Monthly budget** | Low | 122(26.8) |
| | Average | 219(48.1) |
| | Above average | 49(10.8) |
| | Very low | 61(13.4) |
| | High | 4(0.9) |
| | Total | 455(100) |
| **Do you smoke?** | No | 390(85.7) |
| | Occasionally | 33(7.3) |
| | Daily | 32(7) |
| | Total | 455(100) |
| **Do you drink alcoholic drinks?** | No | 444(97.6) |
| | Occasionally | 11(2.4) |
| | Daily | 0 |
| | Total | 455(100) |
| **What is the type of your university?** | Government | 359(78.9) |
| | Foundation / Private | 96(21.1) |
| | Total | 455(100) |
| **Which subject are you currently studying?** | Medical Field | 209(45.9) |
| | Engineering | 165(36.3) |
| | Social Sciences | 72(15.8) |
| | Natural Sciences | 8(1.8) |
| | Total | 454(99.8) |
| | Missing data | 1(0.2) |
| **What is your current academic year in school?** | Preschool | 10(2.2) |
| | Year1 | 97(21.3) |
| | Year2 | 127(27.9) |
| | Year3 | 98(21.5) |
| | Year4 | 96(21.1) |
| | Year5 | 26(5.7) |
| | Year6 | 1(0.2) |
| | Total | 455(100) |

(*Continued*)

**Table 1.** (Continued)

| What is your most recent GPA | Less than 2.5 | 118(25.9) |
|---|---|---|
| | 2.5–2.9 | 170(37.4) |
| | 3.0–3.4 | 118(25.9) |
| | 3.5–4.0 | 49(10.8) |
| | Total | 455(100) |
| **Have you previously failed a year in college?** | No, I have not failed a year. | 373(82) |
| | Yes, I have failed a year / years. | 82(18) |
| | Total | 455(100) |
| **Do you have Family history of mental issues?** | No, we don't. | 380(83.5) |
| | Yes, we have. | 75(16.5) |
| | Total | 455(100) |
| **Do you use psychiatric medication?** | No, I don't. | 415(91.2) |
| | Yes, I do. | 40(8.8) |
| | Total | 455(100) |
| **What do you think is your level of Turkish language?** | Poor | 29(6.4) |
| | Fair | 79(17.4) |
| | Good | 175(38.5) |
| | Very good | 172(37.8) |
| | Total | 455(100) |
| **At present whom do you live with?** | Alone | 43(9.5) |
| | With Parents | 180(39.6) |
| | With housemates | 90(19.8) |
| | With relatives | 11(2.4) |
| | In a dormitory | 131(28.8) |
| | Total | 455(100) |
| **Do you exercise?** | No | 144(31.6) |
| | Yes, occasionally. | 190(41.8) |
| | Yes, less than 90 minutes per week. | 27(5.9) |
| | Yes, more than 90 minutes per week. | 94(20.7) |
| | Total | 455(100) |
| **How would you define the climate between you and your classmates?** | Friendly and relaxed. | 256(56.3) |
| | Competitive and stimulating | 32(7.0) |
| | Competitive and hostile | 29(6.4) |
| | I don't have an opinion yet. | 124(27.3) |
| | I don't have friends. | 13(2.9) |
| | Total | 454(99.8) |
| | Missing | 1(0.2) |

participants, respectively. This distribution underscores the varying degrees of social anxiety's impact on individuals (**Fig 1**).

## Factors associated with social anxiety disorder (SAD)

The inferential analysis yielded significant findings related to SAD. Gender showed statistical significance in both group difference and association tests ($p < 0.0001$). Family history of mental health issues was significant in both group difference tests ($p = 0.001$) and association tests ($p = 0.009$). Exercise habits were also significant in both tests ($p < 0.0001$), as were the type of relationship with classmates ($p < 0.0001$). Smoking status demonstrated statistical

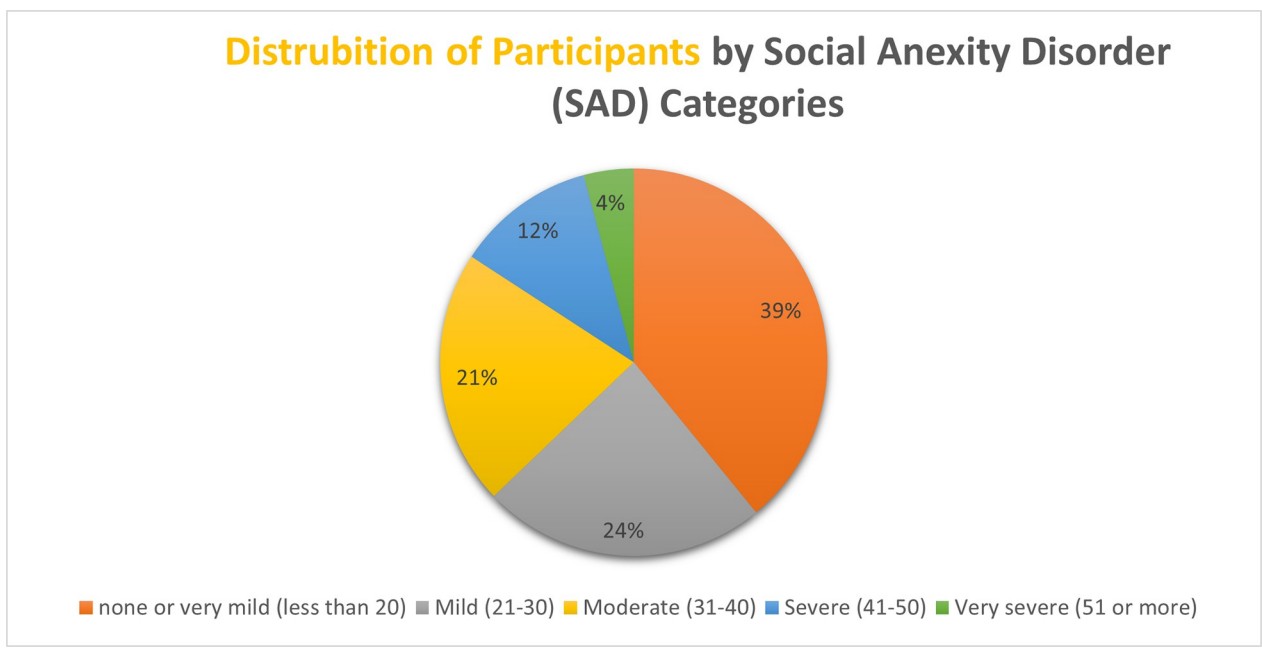

**Fig 1. Prevalence of SPIN categories.**

significance only in the group difference test (p = 0.028), and prior academic failure was significant only in the association test (p = 0.013).

Several factors did not show a statistically significant association with SAD, including age, marital status, monthly budget, alcohol consumption, type of university, subject of study, recent GPA, psychiatric medication usage, level of Turkish language proficiency, and living arrangements (**Table 2**).

### Predictors of social anxiety disorder (SAD)

The multiple regression analysis aimed to predict Social Phobia Inventory (SPIN) scores based on six independent variables: gender, smoking status, previous failure in university, family history of mental health issues, exercise frequency, and type of relationships with classmates. The model significantly predicted SPIN scores (F (6, 447) = 13.165, p < 0.001), explaining 15% of the variance with an adjusted $R^2$ of 14%. Gender, previous failures, family history, exercise, and relationships with classmates were all significant predictors, while smoking status was not. This indicates that gender, family history, exercise, and social relationships play important roles in the severity of social anxiety symptoms, whereas smoking does not have a statistically significant impact (**Table 3**).

### Factors relating to the presence/absence of social anxiety disorder (SAD)

**Table 4** shows the results of the predictors for SAD according to the logistic regression model. The model shows females are found to have a higher probability of SAD compared to males (OR = 1.976, 95% CI = 1.209–3.228, p = 0.007), individuals engaging in more than 90 minutes of exercise per week are less likely to have SAD compared to those who do not exercise (OR = 0.383, 95% CI = 0.207–0.708, p = 0.002). Occasional smokers are shown to have a lower risk of SAD compared to non-smokers (OR = 0.422, 95% CI = 0.183–0.973, p = 0.043). Furthermore, individuals who are undecided about their relationships with classmates are at a

**Table 2. Factors associated with Social Anxiety Disorder (SAD).**

| | | N (%) | SPIN Score. | Group Difference Test | Association Test |
|---|---|---|---|---|---|
| | | **455(100)** | Mean (SD) | *P-value* | *P-value* |
| **Age** | 18–21<br>22–25<br>30–34<br>35 or above<br>Total | 237 (52.1)<br>181 (39.8)<br>27(5.9)<br>10(2.2)<br>455(100) | 26.3(14)<br>25.6(13.2)<br>24.9(15.9)<br>20.4(15.14) | 0.56 | 0.61 |
| **Gender** | Female<br>Male<br>Total | 218 (47.9)<br>237 (52.1)<br>455(100) | 28.89 (14.27)<br>22.96 (12.82) | <0.0001* | <0.0001* |
| **Marital status** | Single<br>Married<br>Widowed<br>Divorced<br>Total | 428 (94.1)<br>24(5.3)<br>1(.2)<br>2(.4)<br>455(100) | 26(13.8)<br>23.2(13.7)<br>20(28.3) | 0.651 | 0.575 |
| **Monthly budget** | Low<br>Average<br>Above average<br>Very low<br>High<br>Total | 122 (26.8)<br>219 (48.1)<br>49(10.8)<br>61(13.4)<br>4(.9)<br>455(100) | 27.7(13.5)<br>26.2(14)<br>25.36 (13.84)<br>24.5 (13.58)<br>23(20.34) | 0.646 | 0.799 |
| **Do you smoke?** | No Smoker<br>Smoker (Occasionally)<br>Smoker (Daily)<br>Total | 390 (85.7)<br>33(7.3)<br>32(7)<br>455(100) | 26.25 (13.77)<br>19.48 (14.25)<br>26.84 (13.07) | 0.028* | 0.461 |
| **Do you drink alcoholic drinks?** | No<br>Occasionally<br>Daily<br>Total | 444 (97.6)<br>11(2.4)<br>0<br>455(100) | 26(13.8)<br>19(13.17) | 0.134 | 0.446 |
| **What is the type of your university?** | Government<br>Foundation / Private<br>Total | 359 (78.9)<br>96(21.1)<br>455(100) | 25.5(14)<br>26.72(13.4) | 0.439 | 0.207 |
| **Which subject are you currently studying?** | Medical Field<br>Engineering<br>Social Sciences<br>Natural Sciences<br>Total<br>Missing data | 209 (45.9)<br>165 (36.3)<br>72(15.8)<br>8(1.8)<br>454 (99.8)<br>1(0.2) | 26.7(14)<br>26.34(13.3)<br>25.46 (14.24)<br>22.45(14.5) | 0.104 | 0.164 |
| **What is your current academic year in school?** | Preschool<br>Year1<br>Year2<br>Year3<br>Year4<br>Year5<br>Year6<br>Total | 10(2.2)<br>97(21.3)<br>127 (27.9)<br>98(21.5)<br>96(21.1)<br>26(5.7)<br>1(0.2)<br>455(100) | 24.9 (17)<br>25.3 (15)<br>25.3(14)<br>28.4(13.6)<br>24.5(11.7)<br>25.6(15.3) | 0.594 | 0.119 |

(*Continued*)

**Table 2.** (Continued)

| | | N (%) | SPIN Score. | Group Difference Test | Association Test |
|---|---|---|---|---|---|
| | | **455(100)** | **Mean (SD)** | *P-value* | *P-value* |
| **What is your most recent GPA?** | Less than 2.5<br>2.5–2.9<br>3.0–3.4<br>3.5–4.0<br>Total | 118 (25.9)<br>170 (37.4)<br>118 (25.9)<br>49(10.8)<br>455(100) | 26.8(13.9)<br>26 (13.3)<br>25.5 (14.24)<br>22.45 (14.5) | 0.399 | 0.374 |
| **Have you previously failed a year in college?** | No, I have not failed a year.<br>Yes, I have failed a year/ years.<br>Total | 373(82)<br>82(18)<br>455(100) | 25.38(14.2)<br>27.7(12) | 0.103 | 0.013* |
| **Do you have Family history of mental issues?** | No, we don't.<br>Yes, we have.<br>Total | 380 (83.5)<br>75(16.5)<br>455(100) | 24.8(13.5)<br>30.8(14.4) | 0.001* | 0.009* |
| **Do you use psychiatric medication?** | No, I don't.<br>Yes, I do.<br>Total | 415 (91.2)<br>40(8.8)<br>455(100) | 25.5 (13.4)<br>28.8 (17.4) | 0.213 | 0.124 |
| **What do you think is your level of Turkish language?** | Poor<br>Fair<br>Good<br>Very good<br>Total | 29(6.4)<br>79(17.4)<br>175 (38.5)<br>172 (37.8)<br>455(100) | 25.9(14)<br>27.5(15.2)<br>27.4(13.4)<br>23.4(13.4) | 0.0524 | 0.260 |
| **At present who do you live with?** | Alone<br>With Parents<br>With housemates<br>With relatives<br>In a dormitory<br>Total | 43(9.5)<br>180 (39.6)<br>90(19.8)<br>11(2.4)<br>131 (28.8)<br>455(100) | 24 (14.8)<br>26.2 (13.4)<br>25.6 (13.7)<br>24.3 (13.7)<br>26.1 (14.4) | 0.881 | 0.909 |
| **Do you exercise?** | No<br>Yes, occasionally.<br>Yes, less than 90 minutes per week.<br>Yes, more than 90 minutes per week.<br>Total | 144 (31.6)<br>190 (41.8)<br>27(5.9)<br>94(20.7)<br>455(100) | 30.5(13.8)<br>25.1(13.3)<br>24.74(12.9)<br>20.4(13.1) | <0.0001* | <0.0001* |
| **How would you define the climate between you and your classmates?** | Friendly and relaxed.<br>Competitive and stimulating<br>Competitive and hostile<br>I don't have an opinion yet.<br>I don't have friends.<br>Total<br>Missing | 256 (56.3)<br>32(7.0)<br>29(6.4)<br>124 (27.3)<br>13(2.9)<br>454 (99.8)<br>1(0.2) | 23.2(13.32)<br>23.44(10.2)<br>29.2(15.92)<br>30.7(13.9)<br>29.8(12.5) | <0.0001* | <0.0001* |

*Indicates significance at p-value < 0.05.

**Table 3. Predictors of social anxiety disorder (SAD).**

| SPIN-Score | B | 95% CI for B | | SE B | Beta | R2 | P |
|---|---|---|---|---|---|---|---|
| Model | | Lower | Upper | | | .139 | |
| **Constant** | 26.975 | 24.531 | 29.420 | 1.244 | | | .000* |
| Gender | -3.773 | -6.406 | -1.141 | 1.339 | -.136 | | .005* |
| Smoking | -.032 | -2.306 | 2.241 | 1.157 | -.001 | | .978 |
| Previously Failed | 3.067 | -.050 | 6.185 | 1.586 | .085 | | .054 |
| Family History | 4.912 | 1.704 | 8.120 | 1.632 | .132 | | .003* |
| Exercise | -2.384 | -3.534 | -1.233 | .585 | -.187 | | .000* |
| Classmates | 1.942 | 1.081 | 2.804 | .438 | .197 | | .000* |

The multiple regression model. B = un-standardized regression coefficient, CI = confidence interval, SE B = Standard error of coefficient, Beta = Standardized coefficient, R2 = coefficient of determination

*Indicates significance at p-value < 0.05

higher risk of SAD compared to those who perceive their relationships as friendly and relaxed (OR = 2.298, 95% CI = 1.367–3.865, p = 0.002).

The model was highly significant ($\chi^2$ (29) = 69.934, p < 0.001) with good fit (Hosmer-Lemeshow p = 0.800), explaining 19.4% (Nagelkerke R2) of variance in social phobia and accurately classifying 68.4% of cases.

## Discussion

As Türkiye continues to attract foreign students from around the globe to pursue higher education at its universities, a significant portion of this diverse group aims to engage with the Turkish community. The country's geographical location and its system of university entrance exams, along with its population diversity, make it a chosen destination for foreign students. This study offers insights into this population of undergraduate foreign students, assessing SAD among them and identifying several factors that may be associated with the presence of SAD. As far as we know, no study examined the incidence of SAD in this population. Given their immersion in a new cultural environment, they encounter challenges in adaptation and language acquisition that can affect their psychological and social health. The high levels of SAD symptoms experienced by students may be attributed to the novel life experiences introduced by university enrollment. These include new living arrangements, forming social relationships, and balancing academic commitments with other responsibilities [15]. In addition, while interpersonal communication and speaking in front of others are crucial skills in students' lives, careers, and academic pursuits, today's college students frequently struggle with interpersonal communication issues, with social anxiety being one of the major psychological issues affecting their academic performance and quality of life [32, 33].

Our study demonstrates that approximately 60.8% of students experienced symptoms of SAD. Among them, 4.2% experienced very severe symptoms, while about one-fifth (11.6%) suffered from severe symptoms. Additionally, about two-thirds of students had mild to moderate SAD symptoms (23.7% and 21.3%), respectively. In agreement with our result, a study reported by Wejdan Al-Johani et al., which included medical students in Saudi Arabia, found that the prevalence of SAD was almost 51%. Moreover, 8.21% and 4.21% reported severe and very severe SAD, respectively [14]. Another study in Saudi Arabia, conducted in 2020 by Al-Hazmi et al., had results that aligned with the outcomes of our study. Using the same SPIN questionnaire we used, they found that among 504 medical students, 13.5% of the participants experienced severe to very severe SAD symptoms [34].

**Table 4. Factors relating to the presence/absence of social anxiety disorder (SAD).**

| Variable | OR | 95% CI For B | | P-value |
|---|---|---|---|---|
| | | Lower | Upper | |
| **Gender** | | | | |
| Male | 1(ref) | | | |
| Female | 1.976 | 1.209 | 3.228 | .007* |
| **Monthly budget** | | | | |
| Very low | 1(ref) | | | .646 |
| Low | .826 | .396 | 1.722 | .610 |
| Average | .781 | .390 | 1.562 | .484 |
| Above average | .583 | .231 | 1.470 | .253 |
| High | .200 | .017 | 2.401 | .204 |
| **Do you smoke?** | | | | |
| No | 1(ref) | | | .101 |
| Occasionally | .422 | .183 | .973 | .043* |
| Daily | 1.188 | .506 | 2.790 | .692 |
| **Studying field** | | | | |
| Medical Field | 1(ref) | | | .517 |
| Engineering | 1.435 | .876 | 2.352 | .152 |
| Social Sciences | 1.053 | .566 | 1.959 | .871 |
| Natural Sciences | 1.476 | .269 | 8.092 | .654 |
| **GPA score** | | | | |
| Less than 2.5 | 1(ref) | | | .532 |
| 2.5–2.9 | 1.008 | .576 | 1.763 | .979 |
| 3.0–3.4 | .819 | .453 | 1.478 | .506 |
| 3.5–4.0 | .623 | .290 | 1.336 | .224 |
| **Family history** | | | | |
| No, we don't. | 1(ref) | | | |
| Yes, we have. | 1.488 | .803 | 2.757 | .207 |
| **Psychiatric medication** | | | | |
| No | 1(ref) | | | |
| Yes | 1.019 | .458 | 2.267 | .964 |
| **Turkish language proficiency** | | | | |
| Poor | 1(ref) | | | .399 |
| Fair | 1.518 | .571 | 4.031 | .403 |
| Good | 1.233 | .501 | 3.035 | .649 |
| Very good | .920 | .368 | 2.297 | .858 |
| **Whom do you live with?** | | | | |
| Alone | 1(ref) | | | .537 |
| With Parents | 1.454 | .647 | 3.267 | .365 |
| With housemates | 1.544 | .655 | 3.639 | .321 |
| With relatives | .555 | .123 | 2.497 | .443 |
| In a dormitory | 1.525 | .675 | 3.448 | .311 |
| **Do you exercise?** | | | | |
| No | 1(ref) | | | .013* |
| Yes, occasionally. | .535 | .321 | .891 | .016* |
| Yes, less than 90 minutes per week. | .758 | .282 | 2.039 | .582 |
| Yes, more than 90 minutes per week. | .383 | .207 | .708 | .002* |
| **Classmates' relationship** | | | | |

(*Continued*)

**Table 4.** (Continued)

| Variable | OR | 95% CI For B | | P-value |
|---|---|---|---|---|
| Friendly and relaxed. | 1(ref) | | | .009* |
| Competitive and stimulating | 2.087 | .898 | 4.853 | .087 |
| Competitive and hostile | 2.338 | .927 | 5.897 | .072 |
| I don't have an opinion yet. | 2.298 | 1.367 | 3.865 | .002* |
| I don't have friends | 2.630 | .643 | 10.755 | .178 |

Logistic regression. OR = Odds ratio, 95% CI For B = 95% Confidence Interval for the coefficient (B)

*Indicates significance at p-value < 0.05

Our study's current results indicate a higher prevalence of SAD compared to a previous study conducted in 2020 by Yared Reta et al. in Ethiopia. Their study found that the prevalence of SAD among 293 medicine and health science undergraduate students was 32.8% [33]. Another study that included 523 university students in Sweden investigated the prevalence of SAD. The point prevalence of SAD among the participants was 16.1% [35]. In the United Kingdom, Graham Russell and Steve Shaw conducted research on students studying in higher education; the findings indicated that approximately 10% of students reported marked to severe SAD [36].

However, comparing our findings with those of the mentioned studies is challenging due to the diverse backgrounds of our participants. Our study focused on assessing the prevalence of SAD among foreign students from various faculties, representing different social factors and cultures. The study included participants from over 25 countries encompassing a wide range of university branches, unlike previous studies that were often limited to specific populations, such as medical students. Syrian students accounted for the majority of responses (59.6%). This outcome was expected as it represents the reality shaped by the civil war in Syria and its proximity to Türkiye, driving many Syrian individuals, particularly students, to pursue education in Türkiye. According to the data of the Council of Higher Education (YÖK) in Türkiye, in 2020, the total number of Syrian refugee students reached 37,236 [37]. It is crucial to consider that Syrian refugees in Türkiye experience symptoms of mental disorders such as anxiety, depression, and PTSD [38]. A great risk of poor psychological health for immigrants has been associated with pressures, such as family conflict, discrimination, and the demands of acculturation [39]. A study conducted in Jordan among Syrian refugee youth aged 15–25 years revealed a high prevalence of SAD, with older youth aged 19–25, who are at the typical university age, being more likely to experience SAD compared to younger individuals, likely due to numerous issues and challenges related to war conflicts and immigration, which can affect their social interactions and relationships, leading to various social and psychological problems [40]. Another study performed in Sudan on long-term internally displaced persons (IDPs) highlighted high prevalence rates of SAD among IDPs, with disability and prolonged displacement significantly increasing the risk, in a country that has experienced one of the worst population displacements in the world due to the world's longest-running civil war [41].

Moreover, Egyptian students, constituting the second largest group in our study, represented 11.4% of the population. This observation underscores the ongoing economic challenges faced by the country, as evidenced by the Egyptian pound being rated one of the poorest-performing currencies in 2023 [42]. Despite having the financial means to do so, the trend of Egyptian youth choosing to study in countries like Turkey instead of traditional destinations like the US or Europe suggests a desire to distance themselves from Egypt due to perceived future uncertainties [43].

Moreover, our study revealed a significant association between tense relationships with classmates, the hostile social environment within the university, and the development of SAD. This observation underscores the challenges students face in integrating into Turkish society and forming supportive relationships to create an encouraging study environment. Indeed, there are numerous local community groups in Türkiye, particularly in major cities like Istanbul which can help students establish a sense of community. However, studying at universities located in these large cities presents significant challenges for students due to the high level of competition. Consequently, many students seek education in urban areas with less diversity in both the population and their peers. This is similar to the findings of Malecha et al., who examined the literature on foreign-born nursing students in the United States; their review reported that foreign-born students face many difficulties in language and communication; barriers such as isolation, discrimination, and cultural differences also affect the educational experience [22]. According to a study by Rosellini and colleagues (2013), 78.8% of individuals with SAD were able to identify the presence of an acute or chronic stressor when their SAD first manifested; this finding aligns with our research, suggesting that exposure to acute stressors, such as a stressful university environment, may influence the development of SAD [44]. In cross-sectional studies, there is a consistent positive correlation between social anxiety and SAD diagnosis with increased levels of relational victimization and social exclusion. Additionally, these conditions are linked to having fewer friends and experiencing challenges in establishing new friendships, lower peer acceptance, more negative interactions within close friendships, as well as greater tendencies toward social avoidance and withdrawal [45–48].

The prevalence of SAD was higher in female participants compared to their male counterparts. This finding aligns with the DSM-5 statement that indicates that SAD is more frequent in women [49]. Consistent patterns emerged in a comprehensive study conducted by Dan J. Stein et al. wherein the analysis of data from 28 community surveys within the World Mental Health Survey Initiative similarly underscored a heightened prevalence of SAD among women [50]. Furthermore, other studies from diverse regions across the globe consistently reveal comparable results [14, 51–53]. In a 2023 study conducted in Hong Kong, however, males showed higher rates of SAD than females; this contradicts results from other studies and can be explained by the acknowledged sampling size bias within the dataset, as the female sample was significantly larger than the male sample [54]. We can understand the higher frequency of SAD in women most effectively through a vulnerability-stress perspective. The notion revolves around the idea that varying exposure to psychosocial stressors, combined with increased susceptibility to anxiety in women, is put forward as a contributing factor to the identified sex differences in anxiety disorders [55].

In addition, our research revealed a favorable correlation between engaging in regular exercise and experiencing fewer symptoms of SAD. This result is in line with public consensus and established scientific evidence supporting the positive influence of exercise on overall mental health, as well as its role in preventing and treating mental disorders [56–60]. The positive influence of exercise may be attributed to different mechanisms. Firstly, exercise could function as a diversion from negative thoughts. Additionally, repeated social interaction may play a role in this process. From a physiological perspective, engaging in physical activity can cause a reduction in cortisol, the stress hormone, potentially leading to an improvement in mood [61]. Also, studies propose that exercise not only fosters the growth of new nerve cells but also releases proteins like brain-derived neurotrophic factor, supporting the survival of nerve cells [62, 63].

Another significant finding was the association between the prevalence of SAD and a family history of mental illness, with a one-half fold increase in the risk of SAD in people with positive family history. This finding aligns with a study conducted in Sweden by Maria Tillfors et al.;

where they found that having an affected family member resulted in a two- to three-fold increase in the risk of SAD [64]. Furthermore, heredity has been identified as a predisposing factor for SAD, and genetic studies have provided strong evidence that at least some genetic component contributes to the development of the disorder [49, 65].

As shown in the results section, the occasional smokers had a mean SPIN score of 19.48, which is considered to have non to very mild social anxiety. This unexpected finding of the current study also is shown as a statistically significant inverse relationship between occasional smoking and social anxiety rates seen in our logistic regression model, with an odds ratio of 0.422. Several reasons could be suggested to explain this paradoxical discovery. To begin, it is possible that smoking on occasion acts as a coping tool for people who suffer from milder forms of social anxiety. In addition, the social situation in which occasional smoking happens may have an impact on social anxiety levels. Previous studies have revealed that smokers frequently smoke in social contexts to satisfy their desire for greater social engagement. This is a frequent condition in smokers with low nicotine dependence [66, 67]. Occasional smoking is frequently associated with social gatherings, where people may feel more comfortable and at ease in the presence of others [68]. However, both daily smokers and non-smokers experience mild social anxiety symptoms, with mean SPIN scores of 26.84 and 26.25, respectively. Yet, there is no statistically significant relation in our logistic regression model. Additional studies on this topic must be conducted to understand whether social anxiety is one of the reasons people start smoking.

As an expected finding, failing a year in college was found to be associated with mild social anxiety symptoms among international students. This finding highlights the specific obstacles that international students may encounter when navigating academic surroundings and transitioning to new social contexts, which leads to an increased vulnerability to social anxiety symptoms following academic losses.

We investigated the prevalence of SAD among students from various academic disciplines, including medicine, engineering, social sciences, and natural sciences. Our findings did not reveal a significant correlation between SAD and the field of study. This contrasts with previous studies, such as one conducted in Ethiopia, which found that SAD was more prevalent among engineering students compared to those studying social sciences and humanities [10]. Additionally, a study conducted among Swedish students reported that SAD was less common among those in pedagogic programs, which primarily attract future teachers [35]. However, research examining the prevalence of SAD across different academic disciplines remains limited. Most studies have focused primarily on medical students, consistently reporting high rates of SAD within this group [14, 34, 69, 70]. Future research should compare the prevalence of SAD across various academic fields rather than concentrating on a single population. Academic success, regardless of the discipline may be more closely associated with SAD than the specific field of study itself.

In discussing the relationship between monthly budget and SAD, it is notable that 48.1% of the participants subjectively assessed their monthly budget as average according to a 5-point Likert scale, while 26.8% considered their monthly budget to be low. Interestingly, the analysis did not reveal a significant correlation between the different monthly budget categories and the prevalence of SAD. In a study among Nigerian students by Bella and Omigbodun, the data suggested that students with lower monthly allowances might experience higher rates of SAD [71]. However, in agreement with our results, the analysis showed that this apparent relationship was not strong enough to confidently conclude that lower monthly allowances cause higher rates of SAD. Other studies worldwide have also investigated the prevalence of SAD and its relation to monthly budgets or family income and have shown similar results [10, 14, 72]. In contrast, one study found that students with higher incomes had less severe SAD [34].

It is reasonable to infer that, since students generally do not work and earn their own income, their lifestyles and financial needs are relatively similar. This homogeneity in their economic conditions may account for the lack of a significant correlation between monthly budget categories and the prevalence of SAD.

Alcohol consumption was another factor examined in our study. However, only 11 participants reported occasional use, while the majority reported no use of alcohol. As a result, no significant relationships between alcohol consumption and SAD was found in our results. This trend likely reflects the cultural and religious beliefs prevalent among our participants coming from predominantly Arab and Muslim countries. Previous studies on this topic, however, have yielded contradictory results. For instance, a Canadian study found that 24% of school dropouts had social phobia and high-risk behaviors like alcohol dependence [73]. This can be explained by the fact that individuals tend to use substances like alcohol as a coping mechanism to manage fear, boost self-esteem and improve social confidence as suggested by a meta-analysis by Etkin and Wager [74]. Despite some reports indicating significant associations between SAD and alcohol related problems [75, 76], other studies didn't support a relationship between hazardous alcohol use and SAD [71, 77, 78]. While one study found no significant correlation between SAD and alcohol use [71], another study found that peer dynamics and social networks have a more prominent effect on alcohol consumption than social anxiety [77]. Moreover, in a longitudinal survey of 1,920 adults above 18 years, researchers found that those with subthreshold forms of SAD were more vulnerable to alcohol related problems than those with the full blown disorder [78]. These findings can be attributed to the tendency of students with full blown social phobia to avoid feared social situations and thus not using alcohol as a coping mechanism.

## Limitations

Our study, being a cross-sectional study, may not have the ability to establish strong conclusions regarding causation. Therefore, longitudinal studies with larger sample sizes are needed to determine other possible factors associated with SAD. Attempting to contact students through social media platforms restricts access to a subset of students, necessitating a comprehensive approach that ensures outreach for most international students. Additionally, the decision to design the questionnaire in Arabic and English excludes students who are not proficient in either language. Furthermore, our study's selection of explanatory variables was not all theoretically supported. Another limitation is the stigma surrounding mental illness among the students, which may prevent individuals from disclosing their condition.

## Conclusion and recommendations

Our study revealed a significantly high prevalence of SAD among foreign-born undergraduate students in Türkiye. Those with noticeable symptoms of SAD tended to be female and have a family history of mental illnesses. Additionally, studying in a stressed atmosphere and experiencing failure were associated with a higher prevalence of SAD. Interestingly, those who smoked occasionally exhibited better symptoms of SAD. These findings necessitate more extra efforts in recognizing and treating SAD among this population and minimizing the negative consequences of adjusting to new life circumstances. This is crucial to protect the mental health of these students. Further research could explore, for instance, the prevalence of SAD among non-foreign-born and foreign-born students in Türkiye. Moreover, future studies could investigate the relationship between SAD and various variables, including the potential mediating effects of other factors on the experience of SAD.

## Supporting information

**S1 Questionnaire. Study questionnaire.**
(DOCX)

## Acknowledgments

We express our gratitude to all respondents who completed the questionnaire, as well as to the administrators of the international students' online groups and the international student associations, for their invaluable assistance in reaching our population.

## Author Contributions

**Conceptualization:** Lujain Alnemr, Abdelaziz H. Salama, Perihan Torun.

**Data curation:** Lujain Alnemr, Abdelaziz H. Salama, Salma Abdelrazek, Hussein Alfakeer, Mohamed Ali Alkhateeb.

**Formal analysis:** Abdelaziz H. Salama.

**Investigation:** Lujain Alnemr, Abdelaziz H. Salama, Salma Abdelrazek, Hussein Alfakeer, Mohamed Ali Alkhateeb.

**Methodology:** Abdelaziz H. Salama, Hussein Alfakeer.

**Project administration:** Lujain Alnemr, Perihan Torun.

**Supervision:** Perihan Torun.

**Validation:** Lujain Alnemr, Abdelaziz H. Salama.

**Writing – original draft:** Lujain Alnemr, Abdelaziz H. Salama, Salma Abdelrazek.

**Writing – review & editing:** Lujain Alnemr, Abdelaziz H. Salama, Salma Abdelrazek, Hussein Alfakeer, Perihan Torun.

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
