## [Decision Letter · Decision Letter 0]

14 May 2024

PGPH-D-24-00820

Prevalence of Social Anxiety Disorder and its associated factors among Foreign-born Undergraduate Students in Türkiye: A Cross-Sectional Study

Dear Dr. Salama,

Thank you for submitting your manuscript to PLOS Global Public Health. After careful consideration, we feel that it has merit but does not fully meet PLOS Global Public Health’s publication criteria as it currently stands. Therefore, we invite you to submit a revised version of the manuscript that addresses the points raised during the review process.

Please address the reviewers' comments

We look forward to receiving your revised manuscript.

Kind regards,

Ejemai Eboreime, MD, MSc, PhD

Academic Editor

Journal Requirements:

Additional Editor Comments (if provided):

Reviewers' comments:

Reviewer's Responses to Questions

**Comments to the Author**

1. Does this manuscript meet PLOS Global Public Health’s publication criteria? Is the manuscript technically sound, and do the data support the conclusions? The manuscript must describe methodologically and ethically rigorous research with conclusions that are appropriately drawn based on the data presented.

Reviewer #1: Yes

Reviewer #2: Yes

2. Has the statistical analysis been performed appropriately and rigorously?

Reviewer #1: Yes

Reviewer #2: Yes

3. Have the authors made all data underlying the findings in their manuscript fully available (please refer to the Data Availability Statement at the start of the manuscript PDF file)?

Reviewer #1: No

Reviewer #2: Yes

4. Is the manuscript presented in an intelligible fashion and written in standard English?

Reviewer #1: Yes

Reviewer #2: Yes

5. Review Comments to the Author

Reviewer #1: Prevalence of Social Anxiety Disorder and its associated factors among Foreign-born Undergraduate Students in Türkiye: A Cross-Sectional Study

Abstract

You need to add keywords

Introduction

Method

Please provide the number of the ethical approval

You need to add more sample size. I don’t think this is sufficient for the study

Discussion

Please explain more detail why the students experienced symptoms SAD based on the references (line 279)

Please do discussion as well with other factors relating to the presence SAD that not significant

Reviewer #2: Interesting paper

Students from countries with civil unrest- Over 60% of the study participants are Syrians- a look at the immigration status could add some interesting information in relationship to SAD. Also for the remaining study participants from Egypt

Monthly budget- Finances- a piece of critical information- how was the range gotten, probably a word or two on this.

On Social connection-closeness to classmates is good, but what about access to local community groups - (country-specific) within Turkey?

6. PLOS authors have the option to publish the peer review history of their article (what does this mean?). If published, this will include your full peer review and any attached files.

**Do you want your identity to be public for this peer review?** For information about this choice, including consent withdrawal, please see our Privacy Policy.

Reviewer #1: No

Reviewer #2: No

---

## [Editor Report · Decision Letter 1]

11 Jul 2024

Prevalence of Social Anxiety Disorder and its associated factors among Foreign-born Undergraduate Students in Türkiye: A Cross-Sectional Study

PGPH-D-24-00820R1

Dear Dr Salama,

We are pleased to inform you that your manuscript 'Prevalence of Social Anxiety Disorder and its associated factors among Foreign-born Undergraduate Students in Türkiye: A Cross-Sectional Study' has been provisionally accepted for publication in PLOS Global Public Health.

Best regards,

Ejemai Eboreime, MD, MSc, PhD

Academic Editor